# Novel insights into serodiagnosis and epidemiology of *Erysipelothrix rhusiopathiae*, a newly recognized pathogen in muskoxen (*Ovibos moschatus*)

**Fabien Mavrot**[1]*, **Karin Orsel**[1], **Wendy Hutchins**[2], **Layne G. Adams**[3], **Kimberlee Beckmen**[4], **John E. Blake**[5], **Sylvia L. Checkley**[1], **Tracy Davison**[6], **Juliette Di Francesco**[1], **Brett Elkin**[6], **Lisa-Marie Leclerc**[7], **Angela Schneider**[1], **Matilde Tomaselli**[1,8], **Susan J. Kutz**[1]

**1** Faculty of Veterinary Medicine, University of Calgary, Calgary, Alberta, Canada, **2** Cumming School of Medicine, University of Calgary, Calgary, Alberta, Canada, **3** US Geological Survey, Alaska Science Center, Sitca, Alaska, United States of America, **4** Alaska Department of Fish and Game, Juneau, Alaska, United States of America, **5** University of Fairbanks, Fairbanks, Alaska, United States of America, **6** Government of the Northwest Territories, Yellowknife, Northwest Territories, Canada, **7** Government of Nunavut, Iqaluit, Nunavut, Canada, **8** Canadian High Arctic Research Station, Polar Knowledge Canada, Ottawa, Nunavut, Canada

* fabien.mavrot@ucalgary.ca

**Data Availability Statement:** All relevant data are within the manuscript and its Supporting Information files.

## Abstract

### Background

Muskoxen are a key species of Arctic ecosystems and are important for food security and socio-economic well-being of many Indigenous communities in the Arctic and Subarctic. Between 2009 and 2014, the bacterium *Erysipelothrix rhusiopathiae* was isolated for the first time in this species in association with multiple mortality events in Canada and Alaska, raising questions regarding the spatiotemporal occurrence of the pathogen and its potential impact on muskox populations.

### Materials and methods

We adapted a commercial porcine *E. rhusiopathiae* enzyme-linked immunosorbent assay to test 958 blood samples that were collected from muskoxen from seven regions in Alaska and the Canadian Arctic between 1976 and 2017. The cut-off between negative and positive results was established using mixture-distribution analysis, a data-driven approach. Based on 818 samples for which a serological status could be determined and with complete information, we calculated trends in sample seroprevalences in population time-series and compared them with population trends in the investigated regions.

### Results

Overall, 219/818 (27.8%, 95% Confidence Interval: 24.7–31.0) samples were classified as positive for exposure to *E. rhusiopathiae*. There were large variations between years and

**Funding:** The following sources supported this work Morris Animal Foundation: D18ZO-407 Natural Sciences and Engineering Research Council Discovery Grant - RGPIN/04171-2014 Natural Sciences and Engineering Research Council Northern Supplement - RGPNS/316244-2014 ArcticNet: 1.5 Muskoxen Polar Knowledge Canada: NST-1718-0015 University of Calgary Eyes High postdoctoral program Shikar Safari Club International Foundation: RSO 1052116

**Competing interests:** The authors have declared that no competing interests exist.

regions. Seropositive animals were found among the earliest serum samples tested; 1976 in Alaska and 1991 in Canada.

In Alaskan muskoxen, sample seroprevalence increased after 2000 and, in two regions, peak seroprevalences occurred simultaneously with population declines. In one of these regions, concurrent unusual mortalities were observed and *E. rhusiopathiae* was isolated from muskox carcasses. In Canada, there was an increase in sample seroprevalence in two muskox populations following known mortality events that had been attributed to *E. rhusiopathiae*.

## Conclusion

Our results indicate widespread exposure of muskoxen to *E. rhusiopathiae* in western Canada and Alaska. Although not new to the Arctic, we documented an increased exposure to the pathogen in several regions concurrent with population declines. Understanding causes for the apparent increased occurrence of this pathogen and its association with large scale mortality events for muskoxen is critical to evaluate the implications for wildlife and wildlife-dependent human populations in the Arctic.

## Introduction

Muskoxen (*Ovibos moschatus*) are distributed across the circumarctic regions of the world [1]. They play an important role in Arctic ecosystems [2–4], are a major source of food and income, and a part of the cultural heritage for many northern Indigenous peoples [5,6]. Nearly extirpated across most of their historical range in the early 20[th] century, drastic conservation measures, including hunting moratorium and bans, as well as translocations, resulted in widespread population recovery by the end of the century [7,8]. In recent years, however, the two largest populations, those on Banks and Victoria Islands, Northwest Territories and Nunavut, Canada, have undergone substantial population declines. The Banks Island population declined from 69,000 to 14,000 between 2001 and 2014 [9]. On northwest Victoria Island, the population dropped from 19,000 in 2001 to 11,000 in 2015 [10]. On the rest of the island, estimates indicated a decrease from 24,000 animals in 1992–94 to 10,000 in 2013–14 [11,12].

Infectious diseases have been identified as a potential threat to wildlife populations globally [13–15]. In muskoxen, multifactorial causes, including diseases and mineral deficiencies, were implicated as causes for the decline of a reintroduced Alaskan population [16]. On Victoria Island, the ongoing decline of muskoxen is concomitant to the apparent emergence or increased occurrence of multiple pathogens and disease syndromes [17–20]. Given the taxonomic uniqueness of muskoxen, and their limited genetic diversity [21,22], which may influence their resilience to diseases [23], it is important to understand the potential role of infectious diseases in their population dynamics and conservation.

*Erysipelothrix rhusiopathiae* is a gram-positive, opportunistic and zoonotic bacterium commonly identified in domestic pigs and poultry, but which can infect a wide range of species, including wild animals [24]. In North American wildlife, sporadic isolation of the bacterium has been previously reported in American bison (*Bison bison*), moose (*Alces alces*), pronghorn antelope (*Antilocapra americana*), white-tailed deer (*Odocoileus virginianus*), and wolf (*Canis lupus*) [25,26]. More recently, *E. rhusiopathiae* has been reported for the first time as a mortality cause in muskoxen between 2010–2013 [27], and has subsequentially been considered as a

potential public health concern in the area [28]. A single genotype of *E. rhusiopathiae* was implicated as the cause of death during multiple muskoxen die-offs in the declining populations of Banks and Victoria Islands in the Northwest Territories and Nunavut, Canada [24]. Subsequently, multiple different genotypes were isolated from carcasses of muskoxen in Alaska, as well as woodland caribou (*Rangifer tarandus caribou)* and moose in Canada, during periods of unusually high mortality of these species [24]. The bacterium has also recently been implicated as the causative agent of a disease syndrome in Pribilof arctic foxes (*Alopex lagopus pribilofensis*) in Alaska [29]. The apparent emergence of *E. rhusiopathiae* as an etiological agent of disease or mortality across a broad host range and spatial scale in temperate and Arctic North America raised questions regarding its historical occurrence and its possible role in the declining health of several muskox populations documented in Canada and Alaska [1,16,19].

The objectives of this study were to develop a species-specific diagnostic serological tool to detect exposure to *E. rhusiopathiae* in muskoxen, describe spatiotemporal trends of seroprevalence to *E. rhusiopathiae* in different muskox populations, and assess seroprevalence relative to known mortality events and population trends in North American muskoxen.

## Materials and methods

### Sample collection

We obtained frozen serum samples or blood on Nobuto filter paper (FP) strips (Toyo Roshi Kaisha, Ltd., Tokyo; Japan; Advantec MFS Inc., Dublin, California, USA distributor) collected between 1976 and 2017 from muskoxen in four regions in Alaska and three regions in Canada (Fig 1 and Table 1). Regions were determined by topographic features for Canada (islands versus mainland) and, for Alaska, by adapting the official Game Management Unit delimitation [30]. For sera, samples were collected during translocation and radio-collaring programs. Whole blood was collected in serum tubes and was kept cool until the serum could be separated from the blood clot by centrifugation within 24 hours of collection. The FP samples were collected as part of hunter-based sampling programs or commercial muskox harvests in Canada [31]. Filter papers were dipped in blood (typically from the jugular or femoral veins or heart) of recently deceased animals, frozen immediately after collection and sent to the University of Calgary where they were processed following the protocol described by Curry et al. [32] to obtain eluates with an estimated dilution of 1:10. All serum and FP samples were stored at -20°C until testing. All sampled muskoxen were free-ranging animals living in remote habitat with no contact with domestic animals. Details on sample collected and their serostatus are given in the supplementary material.

The following wildlife sampling permits and ethic approvals were obtained for this study:

For Alaska: "ADF&G ACUC Approved Protocols: 04–011, 06–08, 08–02, 2010–03, 2010-10R, 2011–012, 2012–04, 2013–18 and US Geological Survey Approved Protocol: 2009–01"

For Canada: "Department of Environment of the Government of Nunavut: 2013–035, 2014–053, 2015–068 and 2016–058" and "Department of Natural Resources of the Government of the Northwest Territories: WL002097, WL002112, WL002853, WL003091, WL500098, WL500158, WL500257,WL005627, WL005761, WL500018, WL 5004469".

For all samples: "University of Calgary Animal Care and Use Permit (AC13-0121)."

### Sample analyses

We modified an ELISA developed by Giménez-Lirola et al. [36]. This ELISA is based on a recombinant polypeptide antigen SpaA (rSpaA415), which has been shown to give reliable results and to be specific to the bacterium *E. rhusiopathiae* with no cross-reaction documented

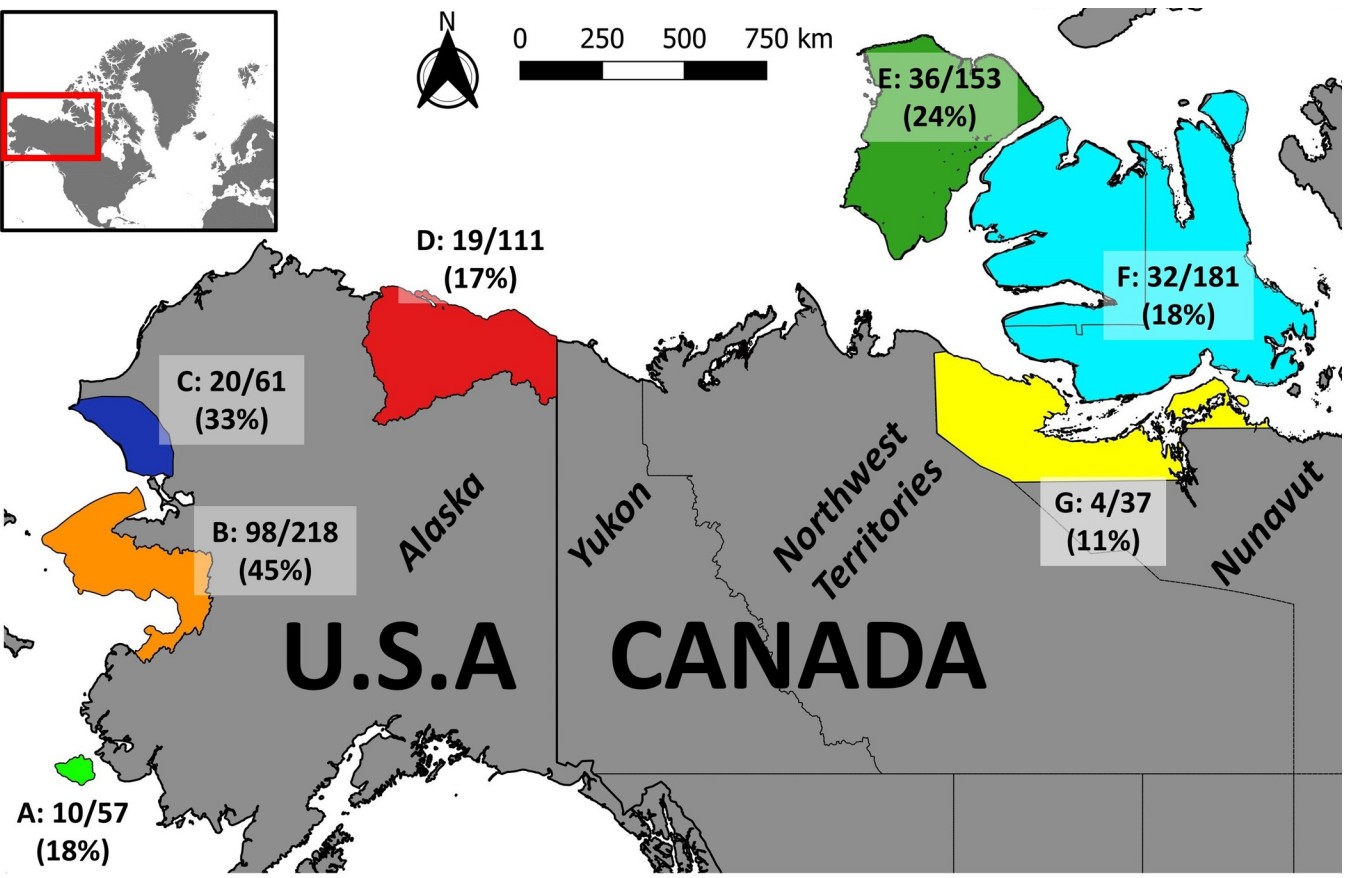

**Fig 1. Study area and origin of samples.** Regions of origin of 818 muskoxen sampled between 1976 and 2017 and serologically tested for exposure to *Erysipelothrix rhusiopathiae*. For each region, the number of seropositive/number of tested individuals and the percentage of positives are indicated. A: Nunivak Island; B: Game Management Unit (GMU) 22; C: GMU 23; D: GMU 26; E: Banks Island; F: Victoria Island; G: Kitikmeot Mainland. The map was created using country and province boundaries from underlined naturalearthdata.com.

with antibodies directed against closely related bacteria or similar cell surface proteins [36,37]. The rSpaA415 gene was recloned to a different vector to increase expression and solubility [38]. In short, the ELISA protocol was optimized as follows: coating 96 well Maxisorp (Nunc) plates with 50 μL of 1 μg/ml SpaA415 in Phosphate Buffered Saline (PBS), incubating overnight at 4°C; washing (all washing steps were done by adding 350 μL/well with 0.1% Tween 20 in PBS three times); adding 300 μL/well of blocking solution (1% Casein, 0.1% bovine serum albumin, 1% Tween 20 in PBS) for 2hrs at room temperature; washing; adding 50 μL/well of sample (serum or FP eluate) in blocking solution diluted 1/10 in PBS (for serum diluent) or in undiluted blocking solution (for FP samples); incubating one hour at 37°C; washing; adding 50 μL/well of Protein A/G-HRP enzyme conjugate (Pierce/Thermo Fisher, Mississauga) diluted to 1/50,000 in serum diluent; incubating one hour at 37°C; washing; adding 100 μL enhanced tetramethylbenzidine-hydrogen peroxide substrate (Pierce/Thermo Fisher, Mississauga) at room temperature for 15–25 minutes; finally, stopping the reaction by adding 100 μL/well 1.00 Normal Sulfuric Acid at room temperature.

All plates were read immediately at 450nm and sample results expressed as optical density (OD) values. Samples were tested in duplicate and the average OD was recorded for each sample. Samples with an average OD value >0.2 and a discrepancy of >20% between duplicates were repeated.

**Table 1. Seroprevalences to *Erysipelothrix rhusiopathiae* in North American muskoxen.**

| Region | Time period | Positive/ total | Sample prevalence (%) | 95% Confidence interval | Population information | Sources |
|---|---|---|---|---|---|---|
| Nunivak Island | 1976–1986 | 10/57 | 17.5 | 9.2–30.4 | Introduced in 1935. Stable, managed, population since the mid 70's. | [33,34] |
| GMU 22 | 1978–1989 | 9/68 | 13.2 | 6.6–24.1 | Introduced in 1970. Growing population in the 70's and 80's. | [33,34] |
| GMU 22 | 2007–2015 | 89/150 | 59.3 | 51–67.2 | Growing in the early 2000s. Population decline in part of the region (North Seward Peninsula) in 2007–2012. Unusually high mortality in radio-collared adults between 2009–2012. *E. rhusiopathiae* isolated from dead animals in 2011–2012. | [1,24,33] |
| GMU 23 | 2009–2012 | 20/61 | 32.8 | 21.6–46.1 | Introduced in 1970. Rapidly growing until 1998 then stagnant and decreasing in 2007–2012. *E. rhusiopathiae* isolated from dead animals in 2012. | [1,24,33] |
| GMU 26 | 1984–1992 | 4/36 | 11.1 | 3.6–27 | Introduced in 1970. Growing population until the mid-90's. | [33] |
| GMU 26 | 2000–2014 | 15/75 | 20 | 12–31.1 | Declining between 2000 and 2007. Since 2007 stabilized at a lower number. | [16,33] |
| Banks Island | 1991–2012 | 36/153 | 23.5 | 17.2–31.2 | Population growing until early 2000's, decline of over 80% between 2000 and 2015 (attributed first to starvation caused by a severe and widespread winter icing event in 2003/2004 and then to unusually high mortality rates associated with *E. rhusiopathiae* in 2012–2013). | [1,9,27] |
| Victoria Island | 2011–2017 | 37/181 | 20.4 | 15–27.2 | Population growing until early 2000's followed by a decline between 2000–2015. Unusually high mortality rates associated with *E. rhusiopathiae* between 2009–2013. | [1,10,12,19,27] |
| Kitikmeot mainland | 2011–2017 | 4/37 | 10.8 | 3.5–26.4 | Recolonization and expansion after near extirpation in the early 1900's. | [1,35] |

Summary of seroprevalences and population information in seven regions of Alaska and Canada investigated for exposure to *Erysipelothrix rhusiopathiae* between 1976 and 2017.

To standardize ELISA results across different plates, we expressed the result of each sample as the percent positivity (PP) of a benchmark positive control, following the formula $PP = (OD_{sample} - OD_{blank})/(OD_{cont} - OD_{blank})$; where $OD_{sample}$ is the optical density value of the sample, $OD_{cont}$ is the value of the positive control of the plate and $OD_{blank}$ is the value of the blank well of the plate. The positive control used across all plates was a pool of five muskox serum samples from our sample set with OD values close to 1.

## Cut-off determination

To estimate the optimal cut-off value of PP to discriminate between negative and positive samples, we used a mixture-distribution modelling (MDM) approach. This method is commonly used to estimate cut-offs for diagnostic tests both in animals [39–41] and in humans [42,43]. Briefly, we assumed that the test results from our sampled population could be represented as a mixture of two underlying subpopulations, corresponding to negative and positive samples, with Gaussian distributions and distinct parameters (mean and standard deviation). Using maximum likelihood estimation, we determined the parameters of each distribution as well as the optimal cut-off (defined as the intersection between the distributions of negative and positive samples) and used 1,000 bootstrapping iterations to compute confidence intervals (CI) around the cut-off. We considered serum and FP samples as two different sets and thus, two separate cut-offs were determined. The R-code we developed to estimate cut-offs and conduct bootstrapping iterations is provided in the supplementary material (S1 File).

## Sample seroprevalences and population trends

We calculated sample seroprevalences (and binomial proportion confidence intervals) for the entire sample set and for the different muskox populations and time periods. In addition, we used general linear modelling (GLM) to construct trends in seroprevalences. For each region and time period, the probability of a sample to be seropositive was modelled as a binomial outcome using the year as a predictive variable. Since the trend could be non-linear, different polynomial degrees of the predictive variable were used in the models. Akaike Information's Criterion was used for model selection. By small (<2) Criterion differences between models, the less complex one (i.e. smaller polynomial degree) was selected. Trends were not computed between data points more than four years apart or for time periods with less than four consecutive years.

Additionally, we reviewed the available literature on muskox population estimates and status in all the investigated populations and presented existing abundance estimates together with seroprevalences.

All statistical analyses were conducted with R [44] using the package mixtools [45] for MD analysis. The map in Fig 1 was created with QGIS [46] using publicly available shapefiles from naturalearthdata.com for the boundaries of North-America.

## Results

### Samples collected and cut-off determination

A total of 958 individual animals (695 sera and 263 FP) were tested for antibodies to *E. rhusiopathiae* and used to run the MDM models and estimate the cut-offs. Values of PP values ranged from < 0.001 to 3.45 (median = 0.14) for serum samples and from < 0.001 to 4.96 (median = 0.20) for FP samples (Fig 2). Estimated distribution parameters of PP values for serum samples, were mean = 0.09 ± SD 0.07 and mean = 1.04 ±SD 0.82 for the populations of negative and positive individuals, respectively. Estimated distribution parameters for FP

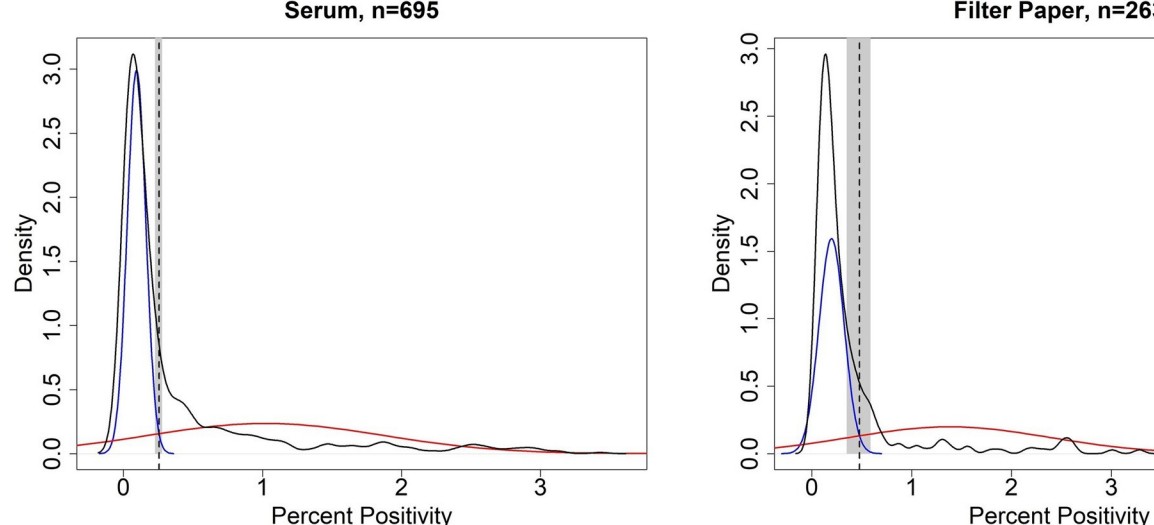

**Fig 2. Mixture distribution model results.** Mixture distribution models for ELISA results of serum and filter paper samples from free-ranging muskoxen tested against *Erysipelothrix rhusiopathiae*. Black curve: frequency distribution of the ELISA results (Percent Positivity) for the datasets (serum or filter paper samples). Blue and red curves: estimated underlying distributions of negative and positive samples, respectively. Dotted vertical lines: cut-off value obtained through mixture distribution analysis. Grey area: 95% Confidence Intervals (CI) computed through 1,000 bootstrap iterations. Optimal cut-offs were estimated at 0.25 (95% CI = 0.23–0.28) and 0.48 (95% CI = 0.35–0.59) for serum and filter paper samples, respectively.

samples were mean = 0.20 ± SD 0.13 and mean = 1.38± SD 1.00 for the populations of negative and positive individuals, respectively. Optimal cut-offs were estimated at 0.25 (95% CI = 0.23–0.28) and 0.44 (95% CI = 0.31–0.60) for serum and FP samples, respectively. Among the samples, sixty-eight samples (7.8%) had PP values within the confidence intervals surrounding the cut-off values, and were considered inconclusive, and excluded from further analysis.

## Sample seroprevalences and population trends

A total of 818 samples (622 sera and 196 FP) with complete information on sampling year and location and which could be classified as positive or negative were included in the analysis. The overall seroprevalence was 227/818 (27.8%, 95% CI = 24.7–31.0). Results from the sample analyses and literature search on muskox population trends are presented in Table 1 and Fig 3. In Alaska, seroprevalence in GMU 22 was higher between 2007–2015 when compared to 1978–1989 (Fisher exact test, p<0.001). A similar but not significant trend was observed in GMU 26 when comparing seroprevalence estimates for 2000–2014 to 1984–1992 (Table 1). In GMU 22 and 26, the highest seroprevalence recorded corresponded to periods just before or during population declines. Additionally, in GMU 22, the peak seroprevalence was concomitant with unusual mortalities and the detection of *E. rhusiopathiae* in muskox carcasses [1,24]. For Victoria Island, although qualitative data on muskox population trends were available in the literature and presented in Table 1, no accurate island-wide estimates were available and thus no population trend was represented for this region in Fig 3

In Canada, Banks Island muskoxen were sampled irregularly between 1991 and 2012. Seroprevalences fluctuated between 2.3% and 41%. The highest seroprevalence occurred concomitantly with a known outbreak of *E. rhusiopathiae*-associated mortalities documented on the island in 2012 [27]. On Victoria Island, where similar *E. rhusiopathiae*-associated mortalities were observed in 2009–2013 [19,27], the sample seroprevalence increased through 2011–2015 from 4.3 to 41.7%. The small sample size and the limited number of sampling years for the Kitikmeot mainland population did not allow for inference on temporal patterns.

## Discussion

### Cut-off determination

In 2009–2013, *E. rhusiopathiae* was for the first time discovered and associated with high mortality rates in muskoxen in the Canadian Arctic Archipelago [27]. Our first step was to assess if the bacterium was new to the Arctic or had historically been present. An important challenge was to ensure that an appropriate diagnostic test was available. Often in wildlife health monitoring, diagnostic tests are adapted from domestic species without calibrating the tests to free-ranging species [47]. Here, we used MDM as a data-driven statistical approach to calibrate our ELISA. This method is widely recognized as a reliable tool to produce cut-offs for diagnostic tests in the absence of a well-established benchmark (e.g. experimental trials) [40–42,48]. However, to effectively use a MDM approach, the investigated dataset must contain both negative and positive samples in sufficient quantities to allow the estimation of the two subpopulations [39]. We assumed that there were seropositive muskoxen in our sample set as samples originated from the same populations where the bacterium *E. rhusiopathiae* had recently been detected [1,27]. Further, the distribution of the PP values was skewed to the right with a long tail, suggesting negative and positive subpopulations [39].

There was a large difference between the means of the distributions of negative and positive samples in our dataset which allowed for a clear determination of cut-offs and indicate that muskoxen can display a strong antibody response to the pathogen (the maximum PP in our set of samples was over 20 times the median value of all tested samples). For both serum and

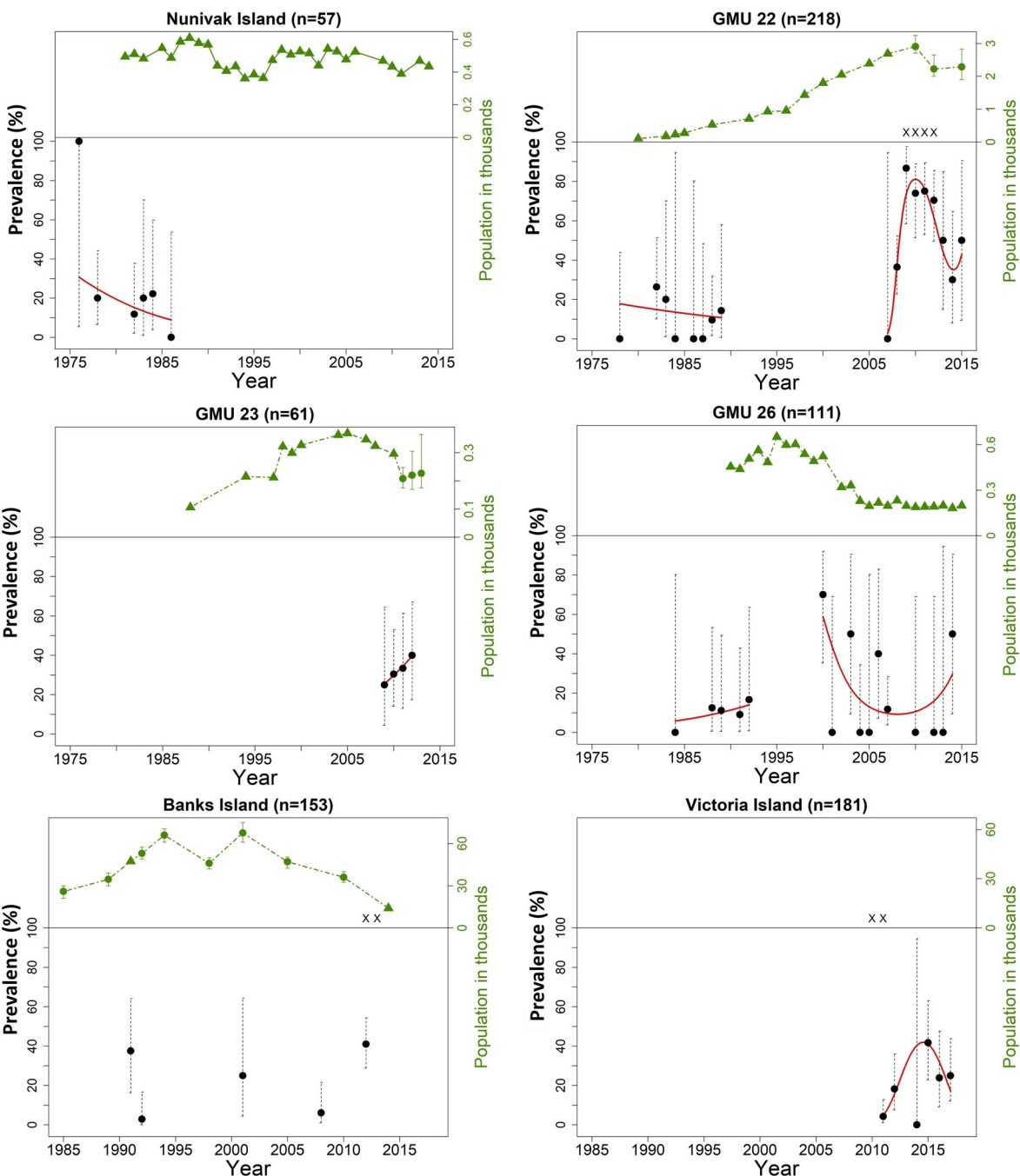

**Fig 3. Time plots of seroprevalences to *Erysipelothrix rhusiopathiae* in North American muskoxen.** Time plots of seroprevalences (black dots) to *Erysipelothrix rhusiopathiae* from a set of 781 muskoxen sampled in six regions of Alaska and Canada. Dotted vertical lines represent exact binomial confidence intervals. Red curves represent trend lines for regions and time periods (see text for the calculations of trends in seroprevalences). Population trends as estimated through aerial surveys are indicated in green in the top part of each plot (with 95% Confidence Intervals represented by green dotted lines) and documented mortality events with detection of *E. rhusiopathiae* are depicted with an X (see Table 1 for references).

FP samples, the estimated distribution of PP values for positive samples was much wider (larger standard deviation) compared to negative samples (Fig 2). This large standard deviation in the distribution of positive samples was previously described by Garnier et al. [39] in a

similar study on ELISA calibration for free-ranging birds. This wider distribution may be explained by the high variability in IgG levels from positive animals corresponding to different immune function and infection history status [49].

## Sample seroprevalence and population trends

Once the cut-offs for our diagnostic tests were estimated, we could effectively classify 818 samples in our dataset as either positive or negative and assess trends in seroprevalences in muskoxen from seven North American regions over more than 40 years. Our results indicate that *E. rhusiopathiae* has been circulating in muskox populations since at least 1976 in Alaska and 1991 in Canada, corresponding to the earliest year for which samples were available. In Canada, the increase in seroprevalence between 2011 and 2015 on Victoria Island, and the high seroprevalence in 2012 on Banks Island, coincided with unusually high mortality rates associated with *E. rhusiopathiae* infection and population declines [1,19]. The severity of the reported mortalities [19,27], the large-scale spread of a unique genotype over a range of 800 km and two islands [24], and the increase in seroprevalence documented on Victoria Island subsequent to the mortalities, suggest an emerging highly infective *E. rhusiopathiae* genotype that caused a widespread seroconversion within a susceptible population. Whether the genotype circulating on these islands in recent years is the same as the one detected serologically in 1991–92 on Banks Island is not known.

On Banks Island, seroprevalence alternated between years with higher values (1991, 2001, 2012) and years with lower values (1992, 2007) The small sample size and the gaps in the time series warrant a cautious interpretation of those results. However, this periodic increase in seroprevalence may suggest a cyclical pattern of outbreaks of *E. rhusiopathiae* similar to those described in other pathogens of free-ranging species [50,51].

The pattern of seropositivity in Alaskan muskoxen differed somewhat from the one documented in Canada. Forde et al. [24] suggested an endemic presence of *E. rhusiopathiae* in Alaskan muskoxen based on phylogenetic analyses that demonstrated multiple different genotypes isolated from the bone marrow of deceased muskoxen in GMU 22 and 23. The hypothesis of endemicity seems to be confirmed by the recurrent low seroprevalence between 1976–1992 in the three Alaskan populations for which archived samples were available prior to 2000. However, in two regions (GMU 22 and 26), yearly seroprevalences over 50% documented after 2000 indicate a possible increase inexposure to *E. rhusiopathiae* in these regions.

Although historically present in muskox populations since at least the 1970s, our results suggest that the seroprevalence of *E. rhusiopathiae*. has increased in some North American muskox populations coincident with observed mortality events and population level declines. The emergence or invasion of new, more pathogenic genotypes could explain this increase. However, while this hypothesis may hold for Victoria and Banks Islands, whole genome sequencing by Forde et al. [24] refutes a hypothesis of a single emerging genotype of *E. rhusiopathiae* in Alaska. In this latter case, mechanisms such as host density-dependent pathogen abundance [52] and stress [53–55] may facilitate negative outcomes of infections with *E. rhusiopathiae*, and could be a contributing factor for muskoxen as well. The Arctic is experiencing rapid changes [56–58], which may increase stress on muskoxen through a variety of biotic and abiotic mechanisms. Possible stressors include extreme weather, changes in predator numbers and occurrence, human disturbance, alterations in plant diversity and phenology, or in mineral availability, and the emergence of other pathogens [59]. Changes in the environment may also modify the ability of pathogens to survive and persist outside the host [27].

Finally, the impact of *E. rhusiopathiae* at the population level remains to be assessed. Here, the documented association between an increase in seroprevalence and population declines/

die-offs alone does not establish causality. Additionally, in our dataset, high seroprevalence against *E. rhusiopathiae* did not always fit known mortality events or population declines. This can be explained by low observation pressure and underreporting of mortalities [19], and also by the fact that the drivers of muskox population dynamics are likely to be more complex Furthermore, the regions used in this study were delimited as epidemiological units consistent with the current knowledge on muskox populations but are nonetheless wide and do not take into account possible subpopulations and heterogeneity in sample collection within each region (e.g. hunter-collected samples clustered around communities). Although not feasible with the data presented here, an analysis of seroprevalences and population trends at a finer spatial resolution might might provide new insights on the association between *E. rhusiopathiae* and muskox population dynamics. Nevertheless, the extent of the 2009–2013 outbreaks on Banks and Victoria Islands and the high number of reported mortalities [27], given that muskox mortalities are substantially under-reported [19], highlight the potential high death toll that *E. rhusiopathiae* can exert on muskox populations. Moreover, diseases might have more pernicious effects on a host population than just direct mortality. Preece et al. [60] noted that pathogens can also have bottom-up negative effects (e.g. reduced fertility) or indirect negative effects such as changes in the population structure, which are more difficult to measure and contribute to the frequent underestimation of the true impact of diseases on wildlife populations. This, together with the data presented here raises the question of whether *E. rhusiopathiae* is not a contributor in the decline of some muskox populations in Canada and Alaska.

## Conclusion

We successfully adapted an ELISA to test for exposure to *E. rhusiopathiae* in free-ranging muskoxen and used it to document the widespread historical exposure in several North American populations toa pathogen that was until recently not known to infect this species. The MDM approach allowed us to estimate an optimal cut-off value for the ELISA without using a set of known-positive and known-negative animals. As species-specific cut-offs are increasingly recognized as necessary in diagnostic testing, this statistical approach is of particular interest to improve diagnostic test accuracy in free ranging-species for which experimental infection trials are not easily feasible. Archival samples were critical for these analyses to understand historical seroprevalence. Our data suggest that in some populations, exposure to *E. rhusiopathiae* has increased concomitantly with observed population declines. Our data also highlight different epidemiological patterns across the investigated regions. Data limitations include many factors such as the lack of knowledge on antibody persistence, small sample sizes, bias due to opportunistic sampling, and missing information on the age and sex of exposed animals. The novel knowledge presented here will be enhanced through further monitoring efforts and the use of complementary, multidisciplinary techniques are warranted to overcome the many limitations linked with health surveillance efforts conducted in remote free-ranging animal populations (Given the need to understand mechanisms that may drive population trends of muskoxen in the Arctic, the importance of muskoxen as a source of food and income for Indigenous communities, as well as the zoonotic potential of *E. rhusiopathiae* documented for other animal species, an enhanced understanding of the epidemiology of this pathogen in a rapidly changing Arctic ecosystem is needed.

## Supporting information

**S1 File. R-code for mixture distribution modeling and bootstrapping to determine optimal cut-off in a set of ELISA results.**
(TXT)

**S1 Data. Csv file with the percent positivity values used to run the mixture distribution modeling and bootstrapping R-code.**
(CSV)

**S2 Data. Excel file with information (location, year, serological status) on all sampled animals included in the study results.**
(XLSX)

## Acknowledgments

We thank the hunters and biologists involved in the sampling as well as the Hunters and Trappers Committees of the communities of Cambridge Bay, Kugluktuk, Sachs Harbor and Ulukhaktok for their support. Many thanks also go to all the volunteers, students and staff members who contributed to sample collection and/or processing and analyses, in particular Russell Akeeagok, Stephen Arthur, Patty DelVecchio, Tony Gorn, Letty Hughes, Elizabeth Lenart, Allen Niptanatiak, Lincoln Parrett, Harry Reynolds, Patricia Reynolds, and James Wang.

We thank Canada North Outfitting the Nunavut Harvester Support Program, the Nunavut General Monitoring Plan, the Governments of NWT and Nunavut, University of Alaska Fairbanks, US Geological Survey, and the US National Park Service for their logistical support. Any use of trade, firm, or product names is for descriptive purposes only and does not imply endorsement by the U.S. or Canadian Government.

## Author Contributions

**Conceptualization:** Fabien Mavrot, Susan J. Kutz.

**Data curation:** Fabien Mavrot, Juliette Di Francesco, Matilde Tomaselli, Susan J. Kutz.

**Formal analysis:** Fabien Mavrot, Karin Orsel, Sylvia L. Checkley, Susan J. Kutz.

**Funding acquisition:** Susan J. Kutz.

**Investigation:** Fabien Mavrot, Angela Schneider, Susan J. Kutz.

**Methodology:** Fabien Mavrot, Karin Orsel, Wendy Hutchins, John E. Blake, Sylvia L. Checkley, Angela Schneider, Susan J. Kutz.

**Project administration:** Fabien Mavrot, Susan J. Kutz.

**Resources:** Fabien Mavrot, Layne G. Adams, Kimberlee Beckmen, Tracy Davison, Juliette Di Francesco, Brett Elkin, Lisa-Marie Leclerc, Matilde Tomaselli, Susan J. Kutz.

**Software:** Fabien Mavrot.

**Supervision:** Susan J. Kutz.

**Visualization:** Fabien Mavrot.

**Writing – original draft:** Fabien Mavrot, Susan J. Kutz.

**Writing – review & editing:** Fabien Mavrot, Karin Orsel, Wendy Hutchins, Layne G. Adams, Kimberlee Beckmen, John E. Blake, Sylvia L. Checkley, Tracy Davison, Juliette Di Francesco, Brett Elkin, Lisa-Marie Leclerc, Angela Schneider, Matilde Tomaselli, Susan J. Kutz.

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
