## [Decision Letter · Decision Letter 0]

10 Jan 2020

PONE-D-19-30167

Novel insights into serodiagnosis and epidemiology of Erysipelothrix rhusiopathiae, a newly recognized pathogen in muskoxen (Ovibos moschatus)

PLOS ONE

Dear Mavrot,

Thank you for submitting your manuscript to PLOS ONE. After careful consideration, we feel that it has merit but does not fully meet PLOS ONE’s publication criteria as it currently stands. Therefore, we invite you to submit a revised version of the manuscript that addresses the specific suggestions raised during the review process.

We would appreciate receiving your revised manuscript by Feb 24 2020 11:59PM. To enhance the reproducibility of your results, we recommend that if applicable you deposit your laboratory protocols in protocols.io, where a protocol can be assigned its own identifier (DOI) such that it can be cited independently in the future. For instructions see: http://journals.plos.org/plosone/s/submission-guidelines#loc-laboratory-protocols

We look forward to receiving your revised manuscript.

Kind regards,

Emmanuel Serrano, PhD

Academic Editor

PLOS ONE

2. Thank you very much for including the following ethics statement on the submission details page: "The following wildlife sampling permits and ethic approvals were obtained for this study:

ADF&G ACUC Approved Protocols : 04-011, 06-08, 08-02, 2010-03, 2010-10R, 2011-012, 2012-04, 2013-18;

US Geological Survey Approved Protocol: 2009-01

Department of Environment of the Government of Nunavut: 2013-035, 2014-053, 2015-068 and 2016-058

Department of Natural Resources of the Government of the Northwest Territories: WL002097, WL002112, WL002853, WL003091, WL500098, WL500158,

WL500257,WL005627, WL005761, WL500018, WL 5004469

University of Calgary Animal Care and Use Permit (AC13-0121)." Please also include this ethics statement in the Methods section and clarify which permits or approvals are obtained for which samples from which countries.

a) You may seek permission from the original copyright holder of Figure 1 to publish the content specifically under the CC BY 4.0 license.  

Reviewers' comments:

Reviewer's Responses to Questions

**Comments to the Author**

1. Is the manuscript technically sound, and do the data support the conclusions?

Reviewer #1: Yes

Reviewer #2: Yes

2. Has the statistical analysis been performed appropriately and rigorously? 

Reviewer #1: Yes

Reviewer #2: Yes

3. Have the authors made all data underlying the findings in their manuscript fully available?

Reviewer #1: Yes

Reviewer #2: Yes

4. Is the manuscript presented in an intelligible fashion and written in standard English?

Reviewer #1: Yes

Reviewer #2: Yes

5. Review Comments to the Author

Reviewer #1: The study addresses an important issue, as muskoxen are an important species for many indigenous communities of the Arctic ecosystem and there has been a large population decline observed since 2001 due to Erysipelothrix rhgusiopahiae, having serious impact. The aim of developing a specific diagnostic serological test is of particular importance not only for the present study, but also for its future use in monitoring animal exposure to the pathogen. In addition, the experimental approach may serve as a model for many other species of wildlife that do not have standard diagnostic tests. Hence, the originality of the approach is something important.

Title is specific, the abstract is accurate, the methodology is fully explained and possible to replicate it. Discussion fully explains the results and especially the ones from Banks Island, where a change in pattern is observed, attributed to a cyclical pattern of outbreaks, a well documented view.

I would suggest to add in your conclusions that your approach could be used in oher wildlife species as well in which a large number of samples is available, and there is an aitiological agent recognized to affect the specific population.

Reviewer #2: This is an important paper that pursued the authors’ previous research papers dealing with large scale mortalities caused by infection with Erysipelothrix rhusiopathiae. In this study, the authors investigated seropositivity for E. rhusiopathiae of blood samples that collected from muskoxen from regions in Alaska and Canada during the past 40 years. They concluded that 227 samples (28%) of 818 tested were positive, indicating widespread historical exposure of the pathogen to the animals in the North American populations.

Overall, I agree to the authors’ conclusion and have minor comments.

Comments are as follows:

1. The authors’ conclusion and interpretation of data is based on that like in pigs, humoral immune responses in muskoxen are well induced by infection of E. rhusiopathiae. However, with experimental infections in cattle, I observed that E. rhusiopathiae does not induce strong humoral immune responses and the immunity does not last long. It may be possible in muskoxen as well. The authors should take it into consideration and can discuss this.

2. Do the authors have any information regarding the animals’ habitat environments? Were all the animals completely in the wild or could they get close to humans, or some farmed? This information is important, if the authors have that, please discuss in detail.

3. L55: “Cl” should be spelled out on first appearance.

4. SpaA is not a membrane protein. So, “membrane proteins” should read “cell-wall-associated proteins” or “cell surface-located proteins” or “cell surface proteins”.

5. Please check whether SpaA415 was coated on ELISA plate in PBS? Not in alkaline buffer?

6. Page 14 Discussion L1 and L3: E. rhusiopathiae is not a syndrome or disease. Please correct them.

7. Page 16, second paragraph can be “On Banks Island, several muskox die-offs were attributed to Yersinia pseudotuberculosis in the 1980’s and 1990’s, but in some cases, despite intensive investigation of Y. pseudotuberculosis at the time, macroscopic lesions were not consistent with yersiniosis (pulmonary edema rather than intestinal lesions [46]) and/or Y. pseudotuberculosis was not isolated from carcasses (1996 die-off [47]). As E. rhusiopathiae was, at that time unknown to affect muskoxen and not specifically looked for, this raises the question of whether those cases were, in fact, unrecognized E. rhusiopathiae infections. On Banks Island, seroprevalence alternated between years with higher values (1991, 2001, 2012) and years with lower values (1992, 2007), and the high seroprevalence documented in 1991 was not associated with any visible population decline or mortality events. The reason for the disassociation is unclear. However, this periodic increase in seroprevalence may suggest a cyclical pattern of outbreaks of E. rhusiopathiae similar to those described in other pathogens of free-ranging species [48,49]."

8. Figure 3: Prevalence (%)

9. Figure 3: Please add Population trend data for Victoria Island. If there is no data, it should be clarified in the text.

6. PLOS authors have the option to publish the peer review history of their article (what does this mean?). If published, this will include your full peer review and any attached files.

Reviewer #1: No

Reviewer #2: Yes: Yoshihiro Shimoji

---

## [Author Response · Author response to Decision Letter 0]

27 Mar 2020

Please find below our responses to the comments of the academic editor and the reviewers.

Academic Editor comments: 

1.Please ensure that your manuscript meets PLOS ONE’s style requirements, including those for file naming.

We changed the file names to comply with the journal quidelines and adapted the text accordingly at lines 708-713

2. Thank you very much for including the following ethics statement on the submission details page: “The following wildlife sampling permits and ethic approvals were obtained for this study:

ADF&G ACUC Approved Protocols : 04-011, 06-08, 08-02, 2010-03, 2010-10R, 2011-012, 2012-04, 2013-18;

US Geological Survey Approved Protocol: 2009-01

Department of Environment of the Government of Nunavut: 2013-035, 2014-053, 2015-068 and 2016-058

Department of Natural Resources of the Government of the Northwest Territories: WL002097, WL002112, WL002853, WL003091, WL500098, WL500158,

WL500257,WL005627, WL005761, WL500018, WL 5004469

University of Calgary Animal Care and Use Permit (AC13-0121).” Please also include this ethics statement in the Methods section and clarify which permits or approvals are obtained for which samples from which countries.

We have added the information at line 120-130

a) You may seek permission from the original copyright holder of Figure 1 to publish the content specifically under the CC BY 4.0 license. 

Figure 1 is not a copy of an existing map and was created for the purpose of this publication. The GIS shapefile used to represent the sampling regions was done by the first author with the software QGIS using the free-hand polygon layer creation tool. The shapefiles used to represent the borders of Alaska and northern Canada were obtained from the website https://mapcruzin.com/ and are publicly available under the Creative Commons Attribution Share-Alike 2.0 license. We modified the Material and Method section at line 190-191 to clarify this and we referenced https://mapcruzin.com/ as the source of the layers for the borders. 

To determine each region, we used the exact locations of the sampled animals (not shown in the publication) and for Alaska, we additionally used the Game Management Units map described (among others) in Miller et al. 2017. For clarity purpose, we added a description on how regions were made at line 111-113 and added the relevant reference from Miller et al., 2017. 

Reviewer 1:

I would suggest to add in your conclusions that your approach could be used in other wildlife species as well in which a large number of samples is available, and there is an etiological agent recognized to affect the specific population

We adapted the text as requested at lines 353-357

Reviewer 2:

1. The authors’ conclusion and interpretation of data is based on that like in pigs, humoral immune responses in muskoxen are well induced by infection of E. rhusiopathiae. However, with experimental infections in cattle, I observed that E. rhusiopathiae does not induce strong humoral immune responses and the immunity does not last long. It may be possible in muskoxen as well. The authors should take it into consideration and can discuss this.

We added some discussion points on those topics at lines 265-267 :

“There was a large difference between the means of the distributions of negative and positive samples in our dataset which allowed for a clear determination of cut-offs and indicate that muskoxen can display a strong immune response to the pathogen (the maximum PP in our set of sample was over 20 times the median value of all tested samples).”

and at lines 361-364:

“It must however be noted that the data presented here only reflects apparent seroprevalence in the investigated populations and is limited by many factors such as the lack of knowledge on antibody persistence, small sample sizes, bias due to opportunistic sampling, or missing information on the age and sex of exposed animals.”

2. Do the authors have any information regarding the animals’ habitat environments? Were all the animals completely in the wild or could they get close to humans, or some farmed? This information is important, if the authors have that, please discuss in detail.

We modified the text at line 120 to clarify that the animals are free-ranging and living in remote areas with no contact with domestic animals. 

3. L55: “Cl” should be spelled out on first appearance.

Corrected as requested

4. SpaA is not a membrane protein. So, “membrane proteins” should read “cell-wall-associated proteins” or “cell surface-located proteins” or “cell surface proteins”. 

Changed membrane protein to cell surface protein (line 144)

5. Please check whether SpaA415 was coated on ELISA plate in PBS? Not in alkaline buffer?

Indeed, we used a PBS buffer. The PBS recipe we use gives a final pH of 7.4 which is deemed sufficient for efficiently binding the antigen to Maxisorp plates. This part of the protocol follows exactly the instructions described in Gimenez-Lirola (2012) from which we adapted our ELISA. The ELISA protocol was modified in our study only to extend the range of species that can be tested, we saw no reason to change the buffer used for coating the plates. 

6. Page 14 Discussion L1 and L3: E. rhusiopathiae is not a syndrome or disease. Please correct them.

We modified the text at line 250-252. 

“In 2009-2013, E. rhusiopathiae was for the first time discovered and associated with high mortality rates in muskoxen in the Canadian Arctic Archipelago. Our first step was to assess if the bacterium was new to the Arctic or had historically been present.”

We also modified the text at line 89-92 to avoid the incorrect use of the term “disease syndrome”

“More recently, E. rhusiopathiae has been reported for the first time as a mortality cause in muskoxen between 2010-2013 [27], and has subsequentially been considered as a potential public health concern in the area [28]”

7. Page 16, second paragraph can be “On Banks Island, several muskox die-offs were attributed to Yersinia pseudotuberculosis in the 1980’s and 1990’s, but in some cases, despite intensive investigation of Y. pseudotuberculosis at the time, macroscopic lesions were not consistent with yersiniosis (pulmonary edema rather than intestinal lesions [46]) and/or Y. pseudotuberculosis was not isolated from carcasses (1996 die-off [47]). As E. rhusiopathiae was, at that time unknown to affect muskoxen and not specifically looked for, this raises the question of whether those cases were, in fact, unrecognized E. rhusiopathiae infections. On Banks Island, seroprevalence alternated between years with higher values (1991, 2001, 2012) and years with lower values (1992, 2007), and the high seroprevalence documented in 1991 was not associated with any visible population decline or mortality events. The reason for the disassociation is unclear. However, this periodic increase in seroprevalence may suggest a cyclical pattern of outbreaks of E. rhusiopathiae similar to those described in other pathogens of free-ranging species [48,49]."

We are grateful to reviewer#2 for bringing this part of the text to our attention. After reflecting on the changes proposed by reviewer#2 and discussing it with one of our co-authors (J. Blake) who was involved in the die-off investigation on Banks Island in the 1980s and 1990s, we have decided to remove the first half of the paragraph. (line 287-294).

“On Banks Island, seroprevalence alternated between years with higher values (1991, 2001, 2012) and years with lower values (1992, 2007). The small sample size and the gaps in the time series warrant a cautious interpretation of those results. However, this periodic increase in seroprevalence may suggest a cyclical pattern of outbreaks of E. rhusiopathiae similar to those described in other pathogens of free-ranging species [51,52]. “

8. Figure 3: Prevalence (%)

Corrected as requested

9. Figure 3: Please add Population trend data for Victoria Island. If there is no data, it should be clarified in the text.

Indeed, no reliable trend data are available for Victoria Island. We clarified this at line 227-229.

Additional changes: 

During the revision process, there was an additional internal review required by our USGS colleagues. This, and our final review of the manuscript, has led to some additional minor edits for clarity. None of these edits have changed the findings or interpretations of the manuscript. These include:

Changed ‘strain’ to genotype for consistency throughout the text.

Line 65: changed “human population in the Arctic” to “wildlife-dependent human populations in the Arctic”

Line 130: added: “Details on sample collected and their serostatus are given in the supplementary material.”

Lines 165 and 167: replaced “determine” with “estimate”

Line 221: added “when comparing seroprevalence estimates for 2000-2014 to 1984-1992” to clarify which trend we are referring to.

Line 222-224 : replaced: ”In GMU 22 and 26, the highest seroprevalence corresponded to the muskox population peak just before they started to decline and, in GMU 22, was concomitant with unusual mortalities and the detection of E. rhusiopathiae in muskox carcasses [1,24]” with ” In GMU 22 and 26, the highest seroprevalence recorded corresponded to periods just before or during population declines. Additionally, in GMU 22, the peak seroprevalence was concomitant with unusual mortalities and the detection of E. rhusiopathiae in muskox carcasses [1,24]”. 

Line 244: In the revision we realized that the phrase ‘’ the sampling period of 2011-2017 from 4.3 to 25%’ was a carry over from a much earlier draft of the manuscript and previous analysis where multiple locations had been combined. We had subsequently organized the data in more refined geographic units which made much more biological sense and this is how all the data have been presented throughout, with the exception of this one oversight. The phrase has now been replaced with “2011-2015 from 4.3 to 41.7%”. This correction does not affect the results nor interpretation of the data. 

Line 274: Changed “established” to “estimated” and removed “effectively”. This was done for clarity.

Line 275: added ”as either positive or negative“.

Line 282: Replaced “and the increase in seroprevalence on these islands documented in this work” with “and the increase in seroprevalence documented on Victoria Island subsequent to the mortalities”. This was done as the data for Banks Island is not sufficient to make the statement for both islands.

Line 304-306: Replaced ”However, in two regions (GMU 22 and 26), the increase in seroprevalence since 2000, with the highest seroprevalence occurring concurrently with a peak in the muskox population, suggests that the epidemiology of the pathogen may have changed.” with “However, in two regions (GMU 22 and 26), yearly seroprevalences over 50% documented after 2000 indicate a possible increase in the presence of the pathogen in these regions.” This was done to simplify and clarify the text. 

Line 312-313: Replaced: “has increased in some North American muskox populations in recent years” with “has increased in some North American muskox populations coincident with observed mortality events and population level declines” 

Line 316: To improve clarity, we shortened the section “In this latter case, other mechanisms such as host density-dependent pathogen abundance [53] may play a role in the documented increase in E. rhusiopathiae occurrence. Furthermore, stress has been implicated in facilitating infections with E. rhusiopathiae in multiple species [54–56] and could be a contributing factor for muskoxen as well.” with “In this latter case, mechanisms such as host density-dependent pathogen abundance [53] and stress[54–56] may facilitate negative outcomes of infections with E. rhusiopathiae [54–56] and could be a contributing factor for muskoxen as well.”

Line 329-332: Added “Additionally, in our dataset, high seroprevalence against E. rhusiopathiae did not always fit known mortality events or population declines. This can be explained by low observation pressure and underreporting of mortalities [19] but also by the fact that the drivers of muskox population dynamics are likely to be more complex.” 

Line 331: Replaced “and, in all likelihood, the drivers of muskox population dynamics are more complex.” with “but also by the fact that the drivers of muskox population dynamics are likely to be more complex.”

Line 339: removed “wide”

Line 346: removed “at least”

Line 350: To improve clarity, we replaced “…to establish the widespread historical exposure to a previously unknown Arctic pathogen in several North American populations” with “…to document the widespread historical exposure in several North American populations of a pathogen that was until recently not known to infect this species.” 

Line 357: Replaced “the historical occurrence” with “historical seroprevalence”

Line 370: added: “documented for other animal species,” because, as of yet, there are no documented cases of transmission of E. rhusiopathiae from muskoxen to people. 

References:

Giménez-Lirola, L. G., Xiao, C. T., Halbur, P. G. & Opriessnig, T. Development of a novel fluorescent microbead-based immunoassay and comparison with three enzyme-linked immunoassays for detection of anti-Erysipelothrix spp. IgG antibodies in pigs with known and unknown exposure. Journal of Microbiological Methods 91, 73–79 (2012).

Miller, S. D., Schoen, J. W. & Schwartz, C. C. Trends in brown bear reduction efforts in Alaska, 1980–2017. ursu 28, 135–149 (2017).

---

## [Editor Report · Decision Letter 1]

31 Mar 2020

Novel insights into serodiagnosis and epidemiology of Erysipelothrix rhusiopathiae, a newly recognized pathogen in muskoxen (Ovibos moschatus)

PONE-D-19-30167R1

Dear Dr. Mavrot,

We are pleased to inform you that your manuscript has been judged scientifically suitable for publication and will be formally accepted for publication once it complies with all outstanding technical requirements.

With kind regards,

Emmanuel Serrano, PhD

Academic Editor

PLOS ONE

Additional Editor Comments (optional):

My congratulations!

Emmanuel
---

## [Editor Report · Acceptance letter]

6 Apr 2020

PONE-D-19-30167R1 

Novel insights into serodiagnosis and epidemiology of Erysipelothrix rhusiopathiae, a newly recognized pathogen in muskoxen (Ovibos moschatus) 

Dear Dr. Mavrot:

I am pleased to inform you that your manuscript has been deemed suitable for publication in PLOS ONE. Congratulations! Your manuscript is now with our production department. 

With kind regards,

on behalf of

Dr. Emmanuel Serrano 

Academic Editor

PLOS ONE